# Performance Analysis and Monitoring of Vanadium Redox Flow Battery via Polarization curves

Kannika Onyu [1], Rungsima Yeetsorn [2],*, Jeff Gostick [3] and Saksitt Chitvuttichot [2]

1 Department of Industrial Chemistry, Faculty of Applied Science, King Mongkut's University of Technology North Bangkok, Bangkok 10800, Thailand
2 Materials and Production Engineering, The Sirindhorn International Thai-German Graduate School of Engineering, King Mongkut's University of Technology North Bangkok, Bangkok 10800, Thailand
3 Department of Chemical Engineering, University of Waterloo, Waterloo, ON N2L 3G1, Canada
* Correspondence: rungsima.y@tggs.kmutnb.ac.th; Tel.: +66-2555-2000 (ext. 2921)

**Abstract:** This article proposes the demonstration and deployment of a hand-tailored vanadium redox flow battery test station to investigate the effect of applied voltages on charging performance for electrolyte preparation and the effect of reactant flow rates on the balance of system capacity. Herein, the two different specifications of membranes and a number of electrode layers playing pivotal roles in the discharging characteristics of the VRFB were observed as well. Results indicated that 1.70 V of the charging voltage was suitable, when optimized voltage was considered from charging time, current, and the mole of electrons. The optimized flow rate (10 mL/min) must be controlled since it corresponds to mass transfer and electrolyte diffusion, resulting in reaction ability on electrode surfaces. The number of layers influenced active areas and the diffusion of electrolytes. Nafion 212 provided superior performance to Nafion 117, because it possessed lower ohmic resistance and allowed for easier proton transfer.

**Keywords:** vanadium redox flow battery; energy storage; vanadium electrolyte preparation; electrode; membrane; polarization curve





## 1. Introduction

Ambitious energy storage targets have been announced by international companies as a result of the growing attention to the energy transition as the route of encouraging a green economy. Energy storage is typically applied to store electricity generated by renewable energy devices in huge volumes. The global market size of energy storage is projected to rise at a CAGR of 5.5% during the forecast period of 2021–2026 (GlobeNewswire) [1], creditable to extensive deployment of shifting emphasis towards renewable power, smart grids, and the commercialization of innovative technologies such as solid-state and flow batteries. Redox flow batteries (RFBs) are extensively considered proper for large-scale energy storage [2] because of their attractive features consisting of unlimited capacity, high round trip efficiency, fast responsiveness, depth of discharge, flexible design, and negligible environmental impact [3,4]. RFBs have 85% efficiency and can be utilized as rechargeable batteries or fuel cells. An RFB is superior to a lithium-ion battery in terms of cycle life (>>13,000 cycles), discharge time (1–10 h), energy cost (150–1000 k$/kW), power cost (600–1500 $·kW/h), and capital cost (<<70 $ MW/h cycles) [5,6].

These performances of the redox flow battery are appropriate for stationary energy storage, since the target of a stationary application is anticipated to achieve 0.05 €/kW h of cost, 10,000 cycles of durability, and a lifespan of 20 years [7]. RFBs have many possible applications such as power storage, load balancing, uninterruptible power sources to supply continuous power to facilities, electric vehicles' rapid charging, and stand-alone power systems [8]. The functions of RFBs depend upon the depth of discharge (DOD), and the high DOD in RFBs does not negatively impact the lifespan of RFBs [9]. The

reactions occurring between the two electrolytes typically overwhelms the activity and durability of the batteries. RFBs have been created depending upon the active redox species in the catholyte and anolyte; thus, there are three major types of RFBs, which are the aqueous system, the hybrid aqueous/non-aqueous system, and the non-aqueous system [10]. VRFBs have emerged as applicable candidates to address large-scale energy storage because the electrochemically active reactants are vanadium species in four different oxidation states in both electrolyte solutions. In other words, they have only vanadium as an active element for the anolyte and catholyte. Applying vanadium as the only active species is principally related to the mitigation of contamination problems between the electrolytes, leading to a longer lifespan, since the cross-contamination of negative and positive electrolytes through the cell separator results in coulombic-efficiency losses [11]. Furthermore, the actively soluble species can be kept without the phase change in the electrodes. The large adoption of RFB systems does not imply that Li-ion batteries will disappear from the market, but the two systems will address different features. Even though VRFBs have reached effective commercial fruition in comparison to other RFBs, the commercialization of vanadium flow battery systems has suffered from the high cost of the vanadium compounds. Therefore, researchers have to either store more electricity in the same amount of vanadium through improved chemistry or improved cell and stack designs [12]. The VRFBs are electrochemical energy storage devices that convert chemical energy into electrical energy through reversible oxidation and reduction of the working fluids, characteristically in two soluble redox couples subsisting in external electrolyte containers sized in concordance with application necessities. A VRFB stack, electrolyte storage, and the balance of the plant part are major compositions of a VRFB system [13]. A VRFB stack typically includes endplates, current collectors, bipolar plates, electrodes, membranes, flow frames, and gaskets. The electrolyte storage section possesses an anolyte and a catholyte contained in sealed containers, whereas the balance of the plant part basically consists of a recirculation loop and a battery management system.

Concisely, vanadium electrolytes, which are $V^{2+}$ and $VO_2^+$, are filled into storage tanks for the discharging process, and after the completed discharging process, the $V^{2+}$ and $VO_2^+$ are converted to $V^{3+}$ and $VO^{2+}$, respectively [14]. The $V^{2+}$ and $VO_2^+$ electrolytes can be prepared using alternative energy resources [15]. The electrolyte's concentration and quantity significantly influence the energy density and performance of VRFBs [16]. The relation between the electrolyte concentrations and a VRFB cell can be investigated via Nernst's Equation (1):

$$E_{cell} = E_{cell}^0 - \frac{RT}{F}\left(\ln \frac{[VO_2^+][V^{2+}]}{[VO^{2+}][V^{3+}]}\right) \tag{1}$$

The parameters in the Equation can be defined as follows: $E^0$ is the formal of cell potential; R is the universal gas constant (8.314472 J/mol K); T is the temperature (K); F is Faraday constant (96,485 C/mol); and [V] (M) represents concentrations of the different vanadium species in solution. Moreover, the electrolyte concentration affects energy efficiency, as illustrated in Equation (1). The energy efficiency can be calculated using Equation (2):

$$\varepsilon_{energy} = \varepsilon_{voltage} \times \varepsilon_{coulombic} \tag{2}$$

$\varepsilon_{energy}$ is energy efficiency, $\varepsilon_{voltage}$ is voltage efficiency, and $\varepsilon_{coulombic}$ is coulombic efficiency [17]. The prepared electrolyte solutions' physical and chemical properties represent critical factors for storing the active redox species carrying the electrical energy [14]. This work is interested in understanding the factors affecting electrolyte preparation for further development. A capable pathway to prepare vanadium electrolyte is to dissolve vanadyl sulfate ($VOSO_4$) in sulfuric acid solution, since it promptly dissolves in aqueous solutions. Moreover, the prepared solution provides excellent stability [18]. It is worth noting that sulfuric acid is preferably considered as a suitable solvent; however, this solution system is restricted by the vanadium ion's solubility and stability. This restriction is mightily dependent on the temperature and concentration of the sulfuric acid [19]. A typical method

entails increasing the current while measuring the voltage's evolution during charge and discharge cycles [20]. Plotting voltage vs. current to obtain a polarization curve is another method of analyzing VRFB performance. Accurate polarization characterization can be assessed using a technique that measures changes in cell voltage while the battery is functioning.

The output current plays an important role in determining losses, namely, activation loss, ohmic loss, and concentration loss inside the VRFB, and the losses are influential factors to consider when comparing VRFB performance [21]. The polarization regions, which correspond to the losses, have been considered important parameters in several studies related to materials selection affecting the electrochemical reaction kinetics, internal resistance, mass transport in VRFB, and electrolyte diffusion through porous electrodes [21]. In addition to the importance of electrolyte preparation, voltage losses in VRFBs have to be reduced. It was found that material resistance principally generates losses. The actual voltage can be calculated from the difference between ideal voltage ($E^0$) and voltage losses, including activation loss ($\eta_{act}$), ohmic loss ($\eta_{ohm}$), and mass transport loss ($\eta_{mass}$) [22]. The important materials needed to be investigated and developed to decrease voltage losses are the membrane and electrode. Cation exchange membranes and anion exchange membranes are the two types of membranes that are commonly used [19,23]. The membrane's primary roles are to separate vanadium ions from the cathode and anode sides, as well as to transport protons to complete the electrical circuit [24]. To complete its functions, a membrane should acquire good ion conductivity, high ion selectivity, low permeation rate of vanadium ions, high chemical stability, and high performance [25,26]. Nafion is a perfluorosulfonic acid proton exchange membrane that consists of a hydrophobic part and a hydrophilic part, and it is made by co-polymerizing unsaturated perfluoroalkyl sulfonyl fluoride with tetrafluoroethylene in varying quantities. Nafion has many types that have different equivalent weights (EW) [27]. EW represents grams of dry Nafion per mole of sulfonic acid groups that affect proton transfer and thickness of membrane related to ohmic loss in the polarization curve. An electrode provides an active area for a redox reaction; thus, it should deliver high electrochemical activity, be electrochemically stable in the various potential windows, have high electrical conductivity to increase kinetics in terms of charge transfer rate, and have chemical stability to prevent corrosion [28]. Conventionally, the flow field of a flow battery can be classified into two major types: flow-through and flow-by [6]. A carbon felt used as an electrode is assembled as the flow-through structure, while carbon papers are applied for the flow-by structure. It is worth noting that the carbon felt and carbon paper electrodes can be utilized for the flow-through design. By stacking electrodes in the right frame, the flow-through structure has the ability to increase the electrode surface area, improve electrolyte homogeneity, and decrease pressure drops. The electrodes' large surface areas contribute to the high voltage efficiency of VRFBs [29]. Additionally, compared to the energy loss in the flow-by structure, a pumping system with a flow-through structure exhibits reduced energy loss.

In terms of cell design, two flow patterns—horizontal and vertical—of electrolyte fed into a VRFB stack are briefly described for a basic understanding. Single cells of VRFBs are connected electrically in a series, and electrolytes are injected into the stack in a parallel direction to a stack structure (Figure 1a). This horizontal pattern establishes a parallel flow pattern leading to electrical shunt current that causes corrosion problems, loss of energy efficiency, and nonuniform distribution of current to the cell [30,31]. The vertical pattern (Figure 1b) occurs when single cells are assembled in a series and the electrolytes are vertically fed into the stack. This stacking feature was designed to prevent electrolyte block, resulting in a current bypass, and to protect water electrolysis. Furthermore, the demand for pump power is lower than parallel feeding.

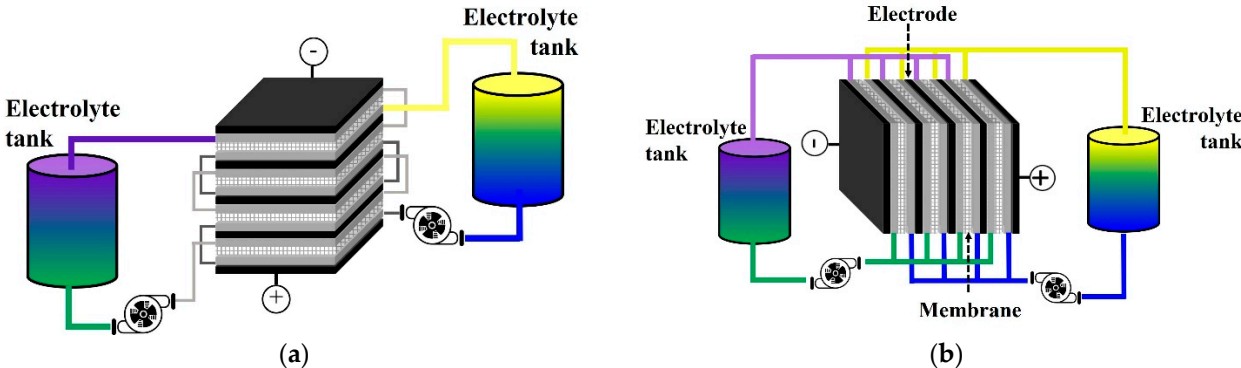

**Figure 1.** Flow pattern of VRFB stack. (**a**) Horizontal VRFB stack; (**b**) Vertical VRFB stack.

In this context, the aim of the present work is to offer an extensive study of the use of polarization curves to investigate the performance of a charging process (electrolyte preparation from vanadyl sulfate ($VOSO_4 \cdot xH_2O$)) and a discharging process. The focus of this study was put primarily on the effects of charging voltage on the performance of the charging step (electrolyte preparation). The performance was evaluated by cell current, moles of electron, and reaction time. In the step of discharging, the electrolyte feeding rate was optimized, since this factor directly relates to the function of carbon electrodes in terms of mass transport. All studies were carried out by operating hand-tailored single-cell VRFBs installed with flow frame design for the carbon paper electrode. Materials selection of main components as an electrode and a membrane was concerned for further design of VRFBs. An important subject for electrode selection is using carbon paper as an electrode instead of carbon felt; thus, the pattern of electrolyte flow related to the number of electrode layers was necessary to investigate. The effects of membrane thickness and the amount of $SO_3^-$ groups in membrane on discharging performance was also studied to be a guideline for membrane selection and design.

## 2. Materials and Methods

### 2.1. Installation of VRFB Single Cell

A VRFB single cell was created to prepare vanadium electrolytes and to investigate battery flow performance. The cell design included two half-cells separated by an ion-exchange membrane. The cell was constructed by a set of components distributed in sandwich layers, consisting of the inclusion of bipolar plates, joints, and carbon paper electrodes, as illustrated in Figure 2. This VRFB cell structure was composed of inlet and outlet channels (Teflon manifold) for the electrolyte and a flow distribution gate, and the vanadium electrolyte was fed by peristaltic pumps. The electrolyte flowed through the manifold, going into the single cells separately. The graphite bipolar plates contributed to the uniform transport of electrolytes and to electron conduction. Teflon flow frames were applied to increase homogenously the electrolyte flow to the electrode. Three-layer carbon paper electrodes (Sigracet SGL 25AA) are active sites where redox reactions occur. The membrane (Nafion Store, Ion Power) was used to separate the anode and cathode and to transfer protons. The sealing structure of VRFB was utilized to prevent internal and external leakages in the cell. In this work, fluorine–rubber gaskets with the proper features were designed for preventing leakages. The components of our VRFB cell were assembled in the structure under compressive pressure in the following sequence: end plate, manifold, current collector, bipolar plate, gasket, flow frame, electrode, gasket, and membrane. Both sides of the cell were assembled as a mirror sequence. After the system was installed as in Figure 2, the leakage test of the cell was conducted. The specifications of the VRFB used in this work are presented in Table 1.

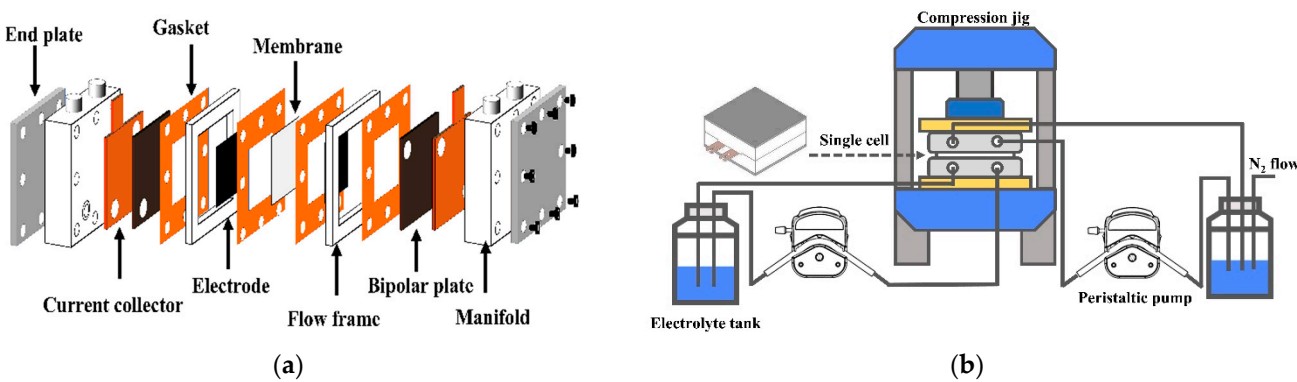

**Figure 2.** (**a**) Diagrammatic assembly drawings of our in-house VRFB cell and (**b**) the VRFB system installation.

**Table 1.** The specifications of the VRFB single cell.

| Specifications | Values |
| --- | --- |
| Single-cell dimension (width × length × height) | 126 mm × 126 mm × 67 mm |
| Active area | 30 mm × 30 mm |
| Temperature range during operation | 23–30 °C |
| Voltage range | 1.2 V–1.6 V |
| Cell current density | 40–100 mA cm$^{-2}$ |
| Maximum current | 4.9 A |
| Compression pressure | 87 psi (0.6 MPa), Force is recommended at 116 psi (0.8 MPa) max. |

### 2.2. Electrolyte Solution Preparation

The experimental approach included the preparation of an electrolyte solution, and this step was started by dissolving 1.5 M vanadium (IV) oxide sulfate hydrate (VOSO$_4$, 99.9%, Alfa Aesar, Lancashire, UK) in 2.6 M sulfuric acid (H$_2$SO$_4$, 95–97%, Merck, Darmstadt, Germany). To achieve good mixing, the mixture of VOSO$_4$ and H$_2$SO$_4$ solutions were stirred until the dark blue color of the electrolyte solution turned to transparent blue. Electrolyte reservoirs were filled with 30 mL of electrolyte solution on the negative side and 60 mL on the positive side. A peristaltic pump (WATSON-MARLOW 120U, Cornwall, UK) with a tube set (Marprene peristaltic pump tubing, 1.6 mm inside diameter and 1.6 mm wall thickness) was used to circulate catholyte and anolyte between reservoirs and the VRFB single cell.

### 2.3. V(II) and V(V) Electrolyte Preparation Via Electrochemical Processes and VRFB Performance Tests

The following step is related to electrolysis to generate vanadium ion species. A potentiostat (BioLogic, SP150) was used to apply an electrical potential to the VRFB in order to generate V(II) and V(V). In this section, constant voltage was optimized by varying the applied voltages between 1.6 V and 2.0 V. Because the electrical potential needed to generate a redox reaction should be higher than the ideal voltage, the voltage should be applied between 1.6 and 2.0 V The catholyte was converted to VO$_2^+$, while the anolyte was converted to V$^{2+}$ [32,33]. Cell voltage and cell current as a function of time were measured to investigate the capability of the electrolyte preparation. The characteristics of I–V curves generated by the discharging process at 100% state of charge were investigated. Different electrolyte feeding rates (5–30 mL/min) were also imposed to improve VRFB performance. According to the assumption regarding the feasibility of using carbon papers as electrodes in a VRFB, the appropriate electrode layer number was assembled for both charging and discharging processes, ranging from 3 to 10 layers. Furthermore, two grades of Nafion

membranes (Nafion 117, 212) were used to investigate proton transfer performance as it relates to redox reactivity.

## 3. Results

### 3.1. The Interaction of Electrolyte Preparation and Discharging Performance

Electrolytes mainly store the active redox species carrying electrical energy, so physical–chemical properties—solubility, stability, ionic conductivity, and viscosity—of prepared solutions signify critical factors for suitable functioning. The vanadium electrolytes are mostly gained from $VOSO_4$ or $V_2O_5$ [34]. Nonetheless, $V_2O_5$ naturally has very low solubility, resulting in a difficult-to-prepare electrolyte solution with a high vanadium concentration directly. This rational restriction leads to efforts to prepare solutions from other vanadium compounds such as $V_2O_3$ and $VCl_3$ [35], but their expensive cost, low solubility, and low stability limit their applications exclusively for large-scale systems. Preparing electrolytes using vanadyl sulfate ($VOSO_4$) as an initial reactant creates $VO^{2+}$, which is responsible for producing $V^{2+}$ and $VO_2^+$ representing the state of charge (SOC) of 100% [35]. This implies that the reaction can be readily applied for a discharging step. This preparation method is suitable for refueling vanadium electrolytes in an electric vehicle tank. However, it is necessary to be concerned about low electrolyte solubility and electrolyte susceptibility oxidation to the air of the species $V^{2+}$ cause the low energy density of VRFB [12]. In terms of utilizing vanadium pentoxide ($V_2O_5$) as the initial reactant for the electrolyte preparation, dissolving vanadium pentoxide in sulfuric acid generates $V^{3.5+}$, which is the combination of $V^{3+}$ and $VO^{2+}$ electrolytes corresponding to a SOC of 0% [19], assuming that the VRFB must be charged before use. This pathway can be applied for EV charging station load, agriculture, and telecommunications integrated with solar cells and/or wind turbines as primary energy resources.

In this study, electrolytic dissolution was used to begin the electrolyte production process utilizing $VO^{2+}$ as a source [35]. Since water is a naturally occurring solvent for electrolytes and is a highly polar solvent with the ability to dissolve ionic salts, using water as a solvent for electrolyte preparation is typically considered. That is due to meaningful differences in the electronegativity of hydrogen and oxygen atoms. Both cation and anion solutes excellently interact with water molecules, bringing about an extended solvation structure that theoretically corresponds to ion hydration [36]. Reactants for preparing electrolytes consist of active species and supporting electrolytes. Supporting electrolytes assist in increasing energy storage and ionic conductivity. As mentioned in the previous part, sulfuric acid has been extensively used as a supporting electrolyte to provide rational solubility (1.5–1.6 M). Note that $VOSO_4$ is adopted as a starting reagent since it possesses higher solubility in an aqueous $H_2SO_4$ solution that is more than 10 times higher than that of $V_2O_5$ [37]. In our work, the electrolyte preparation was begun from $VOSO_4$ in both half-cells, but the reservoir of the cathode side contained twice as much electrolyte solution as the reservoir of the anode side did. A logical prerequisite for imposing the electrolyte volume of the cathode and anode sides is the stoichiometry of the redox reaction occurring in a VRFB cell. One electron is lost during $VO^{2+}$ transformations into $VO_2^+$ during oxidation, while two electrons are accepted during the reduction when $VO^{2+}$ is transformed into $V^{2+}$. That indicates that the $VO_2^+$:$V^{2+}$ ratio is equivalent to 2:1, so in order to establish stoichiometry, the catholyte must be present in greater amounts than the anolyte [32,33]. A flow chart of the electrolyte preparation and discharging performance is shown in Figure 3. For experimental activities, the positive pole was connected as a working electrode, whereas the negative pole was linked with reference and counter electrodes. It should be emphasized that there is a narrowing of the operating temperature range (10 °C–40 °C). It undergoes thermal precipitation at temperatures above 40 °C and dissipation when below 10 °C [37]. Low temperatures are also related to the precipitation of species $V^{2+}$, $V^{3+}$, and $VO^{2+}$ [38]. The electrical potential applied to the VRFB cell generated an oxidation reaction on the positive side and reduction on the negative side. When oxidation occurred, $VO^{2+}$ was transformed into $VO_2^+$. This transformation can be observed

by monitoring the electrolyte's color change from blue to yellow, as presented in Figure 4. The vanadium metal is classified as a transition metal if the oxidation number changes, causing the color change (Equation (3)). In terms of the reaction, the reaction mechanism involved two steps (Equation (4) and (5)) [6]. $VO^{2+}$ turned to be $V^{3+}$ in the first step, and then a blue solution became green. In the second step, $V^{3+}$ accepted the electron, and then $V^{2+}$ was produced. The color of the electrolyte changed from green to purple, as indicated in Figure 3. Electrolytes were stored in an external reservoir [39] and were injected into the VRFB with the same flow rate. The redox reaction occurred on the electrode surfaces, and once it was finished, the products there were moved to the electrolyte reservoir for storage. According to this procedure, the migration and concentration change of positive and negative electrolytes were unaffected by the difference in electrolyte volumes.

$$\text{Cathode } VO^{2+} + H_2O \rightarrow VO_2^+ + 2H^+ + e^- \quad E^0 = 1.00 \text{ V} \tag{3}$$

$$\text{Anode } VO^{2+} + 2H^+ + e^- \rightarrow V^{3+} + H_2O \quad E^0 = 0.34 \text{ V} \tag{4}$$

$$V^{3+} + e^- \rightarrow V^{2+} \quad E^0 = -0.25 \text{ V} \tag{5}$$

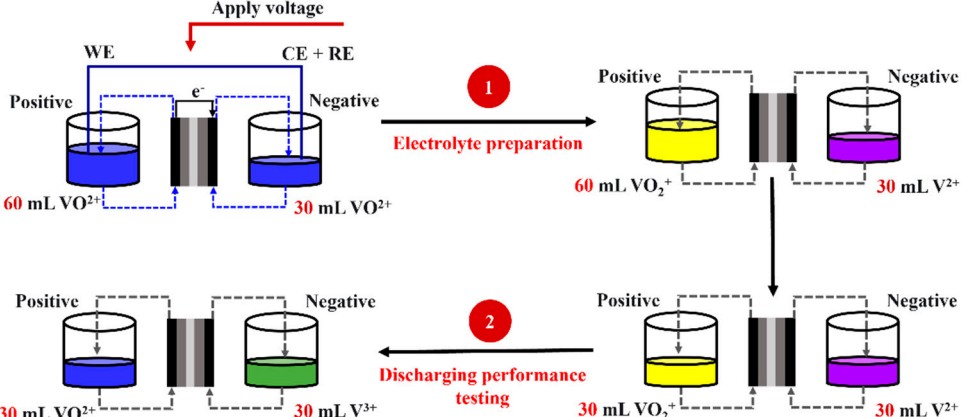

**Figure 3.** Flow chart of electrolyte preparation and discharging performance.

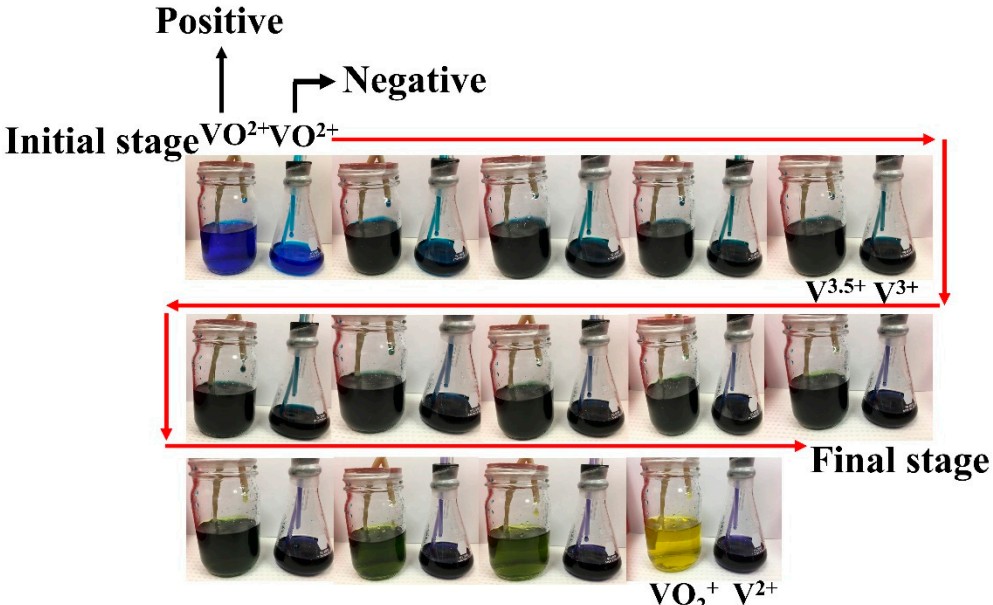

**Figure 4.** The change in electrolyte colors on the positive and negative sides.

The actual operating voltage is a critical factor when charging VRFB or preparing electrolytes because the proper operating voltage (applied voltage) can drive an efficient electrolytic reaction with no side reactions. Because the amount of overpotential is required in addition to the thermodynamic voltage, the charging voltage must be greater than 1.26 V. The charging voltage affects the overall kinetics of the electrochemical reaction caused by charge transfer. If the VRFB is overcharged, hydrogen and oxygen gas may be produced at the negative and positive electrodes. Graphite plates can also be corroded, resulting in the production of carbon dioxide gas [6]. As a result, in this experimental section, the optimal charging voltage was investigated. Figure 5 illustrates the current profiles of the VRFB that were measured as a function of time during the electrolyte preparation process. The VRFB was electrically supplied via chronoamperometry by imposing a certain electrical potential (1.6–2.0 V). The system current measured during the preparation step, the reaction duration, the moles of electrons involved, and the electric charge involved in the redox reaction are the values that were observed to determine the appropriate voltage that was applied to initiate the reaction. When the electrolyte is introduced into the VRFB cell, a concentration gradient will cause the diffusion process to occur from a high concentration to a low concentration. Following that, a redox reaction takes place on the electrode surface, resulting in an electron transfer. The two main factors affecting electron mobility are surface morphology and electrode structure. Electrons technically move to an external circuit or adjacent electrodes via the electron migration process [40].

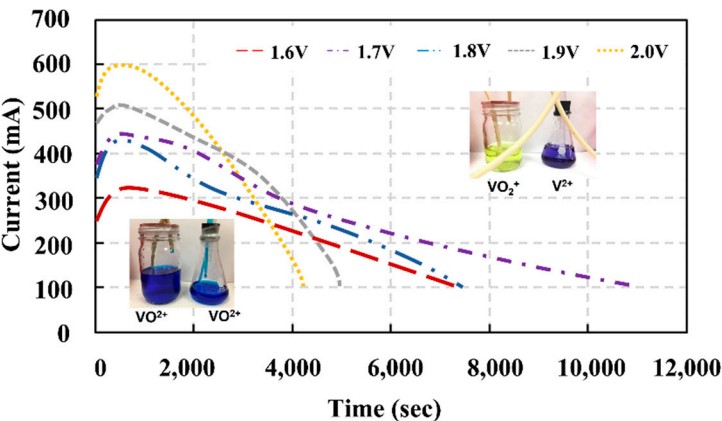

**Figure 5.** The current profile of the VRFB cell as a function of time during the electrolyte preparation period.

The electrical potential was applied to the vanadium electrolyte fed into the VRFB cell, and then the potential energy was transformed to kinetic energy or driving force of electron transfer from the cathode side to the anode side. The higher resulting current monitored during the process indicated higher reaction efficiency. In the next stage, the system current significantly decreased, since the $VO^{2+}$ quantity was significantly reduced after the redox reaction was undergone. The reduction of reactant concentration powerfully affected the productivity of the redox reaction. Figure 6 depicts two criteria for imposing the appropriate potential for electrolyte preparation. The highest current observed during the process is the first criterion. The second pertains to a reaction period calculated from the time the current was reduced to 100 mA. After 100 mA, the current will reach an equilibrium state, implying that the concentration of ion products cannot be increased further. This decrease in current is consistent with the theory in terms of reactant concentration reduction when the redox was carried out for a certain period. Although the cell did not generate the highest current during the redox, applying an electrical potential to the VRFB at 1.7 V provided the most complete reaction possible. According to this scenario, the rate of the current drop is minimal. The moles of electrons determined during redox generated by various applied voltages are shown in Figure 5. Because the high quantity of electron moles

in the electrolytes substantiates the reaction efficiency, it was discovered that applying 1.7 V provided the highest electron moles existing in the system [41].

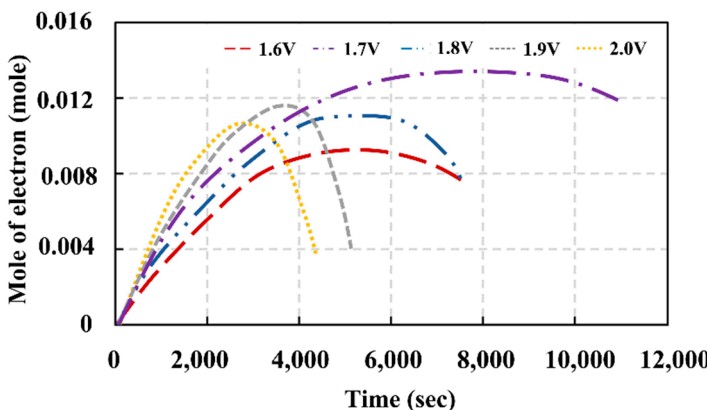

**Figure 6.** The moles of electrons occurring during redox as a function of time during the electrolyte preparation period.

The application of a voltage of 2.0 V appeared to be the best condition for driving the redox reaction; however, the dramatic decrease in current after the peak may be related to a side reaction. The side reaction has a negative impact on electrolyte preparation performance. As shown in Equations (6) and (7), the side reactions caused by overcharging will generate hydrogen and oxygen gases at the negative and positive electrodes, as shown in the Equations (6) and (7). The optimum applied voltage imposed for an electrolyte preparation can prevent the side reaction, as stated in our previous publication [42].

$$2H^+ + e^- \rightarrow H_2 \tag{6}$$

$$2H_2O \rightarrow O_2 + 4H^+ + 4e^- \tag{7}$$

Gas evolution should be avoided because it disrupts electrolyte flow, causing pH changes and increasing cell resistance. Furthermore, oxygen evolution can oxidize the carbon electrode (positive half-cell) [43]. The efficient performance of the electrochemical cell is critical for VRFBs because it is dependent on the reaction kinetics and energy density of a VRFB, both of which are influenced by the concentration of vanadium ions in the electrolytes. The properly applied voltage that resulted in the optimum concentration of vanadium species was concerned with improving performance in this regard. Polarization curves are without a doubt one of the most commonly used tools for assessing the performance of electrochemical devices [40]. The volume of the $VO^{2+}$ solution is halved at this stage due to the redox stoichiometry that occurs during the electric discharge process. The electrolyte preparation is referred to as the VRFB's initial charge. $V^{3+}$ was in the negative electrolyte after this process, while $VO^{2+}$ was in the positive electrolyte. A $VO^{2+}$ was injected into the cathode side to convert the produced $VO_2^+$ solution back to $VO^{2+}$, resulting in discharged VRFB, while the $V^{3+}$ electrolyte was converted to $V^{2+}$ at the anode side. With two different electrolytes prepared by different applied voltages, ascending polarization curves were performed from an open circuit to high electrical loads (Figure 7). Figure 7a shows that these polarization curves (I–V curves) did not exhibit significant voltage drops in a low current density region. Regardless, the VRFB that used electrolytes prepared by 2.0 V of applied voltage resulted in a greater loss than the others in an activation loss region, which is consistent with the experimental results shown in Figures 5 and 6. The curve trend shows that the VRFB with electrolytes prepared by 1.7 V of applied voltages had the lowest voltage drops in medium and high current densities, implying that this VRFB had the lowest ohmic loss and concentration losses. When compared to the other cells, this cell had the highest power (38.46 mW). It is worth noting that the curve is reversible at current densities of around 68.00 mA/cm$^2$, indicating the presence of a complete redox

reaction [33]. The maximum power and voltage efficiency reported in Table 2 provide a better understanding of the effect of initial charge voltage on discharging performance. As shown in Table 2, the VRFB operated with electrolytes charged by 1.7 V of applied voltage outperformed the other cells. This phenomenon could be attributed to stoichiometry effects on the performance of the VRFBs in terms of maximum power and voltage efficiency caused by the completion of the electrochemical reaction during initial charging and the concentration of species in the catholyte and anolyte ($VO_2^+$ and $V^{2+}$) [44]. The circulation of electrolytes in the VRFB batch system led to the observation of a reverse curve in the area of high current density. In this case, the electrolyte products and electrolyte reactants were combined, which caused the reactant concentration to drop, lowering the VRFB current density. However, because the electrolyte reactants were largely transformed into products, this reverse curve reflected the success of the conversion [20].

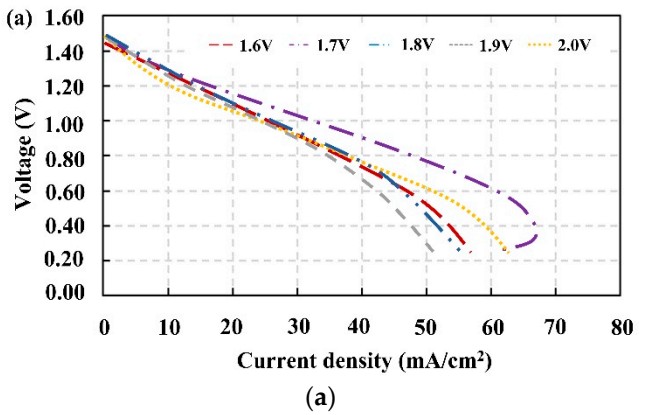 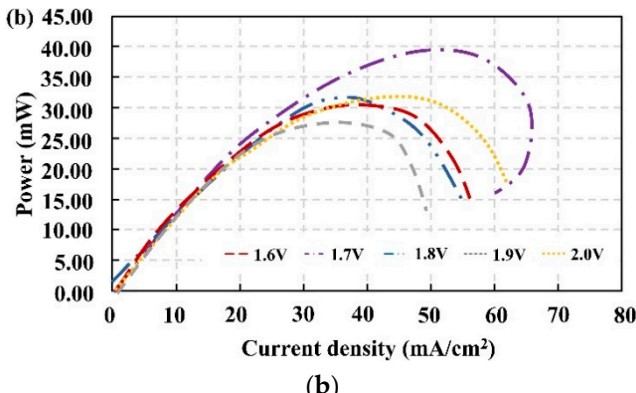

(a)  (b)

**Figure 7.** (**a**) Polarization curves and (**b**) power vs. current density plots obtained from the discharging process following electrolyte preparation with various applied voltages.

**Table 2.** Maximum power and voltage efficiency obtained from polarization curves.

| Applied Voltage | Maximum Power (mW) | Voltage Efficiency (%) |
|---|---|---|
| 1.6 V | 30.28 | 53.45 |
| 1.7 V | 38.46 | 50.46 |
| 1.8 V | 27.40 | 48.38 |
| 1.9 V | 30.80 | 45.64 |
| 2.0 V | 31.27 | 43.57 |

*3.2. Flow Rate's Influence on Discharging Performance*

After being converted into chemical energy, electrical energy is typically stored in external tanks in the form of an electrolyte solution, and electrolyte transport occurs through pipes, with the flow aided by pumps. The flow rate of the electrolyte is directly proportional to the efficiency of the battery's operation [44]. A higher electrolyte flow rate improves the overall performance of VRFBs. A high increase in flow rate, on the other hand, may result in high energy consumption during loading, reducing the VRFB's energy efficiency [31]. As a result, it is critical to determine an appropriate electrolyte flow rate in order to balance system capacity acquisition with efficiency loss [45]. It should be noted that the discharging in this step used charged reactants produced by 1.7 V of the charging voltage.

The electrochemical polarization curves determined during the charging period illustrated that the performance of the VRFBs was increased with an increase in the electrolyte flow rate (Figure 8). The polarization curves show that the ohmic and concentration losses increased when electrolytes were fed at the lowest rate of 5 mL/min. On the other hand, in the areas of activation, ohmic, and concentration losses, the highest rate offered the slower rate of voltage decay. The performance gain at the focus loss was the most striking among these improvements. The concentration gradient between the concentration of

the bulk electrolyte and the concentration of the electrode surface is what is thought to be responsible for the restricted mass transit [44]. As a result, feeding electrolytes into the VRFB cell at an appropriate flow rate can alleviate this limitation and improve the reaction capability on the electrode surface. The lower the flow rate, the lower the mass transport coefficient and the wider the Nernst diffusion layer [11]. The link between the flow rate and the current produced on the electrode is shown in Equation (8) [46]. Along with affecting concentration loss, the electrolyte flow rate also has an impact on the liming current density. Electrolyte flow rates can be split into two categories: the flow rate of the electrolyte reactant diffusing through porous electrodes, and the flow rate of the electrolyte reactant flowing into a flow battery. The electrode structure and design affect the electrolyte diffusion through porous electrodes. During operation, the electrolyte flow rate going into the battery can be changed.

$$\mathrm{i} = 0.9783 \frac{\mathrm{nFD_c}}{\mathrm{L}} \int_0^{\mathrm{L}} \left( \frac{\mathrm{u_f}}{\mathrm{hDx}} \right)^{1/3} \mathrm{dx} \tag{8}$$

where L is the length of the flow channel, $\mathrm{u_f}$ is the averaged electrolyte flow velocity along the flow channel, h is the distance between the electrode and one flat plate, c is the bulk concentration, D is the diffusion coefficient, F is Faraday's constant, n is the number of elementary charges transferred by each ion, and i is the current.

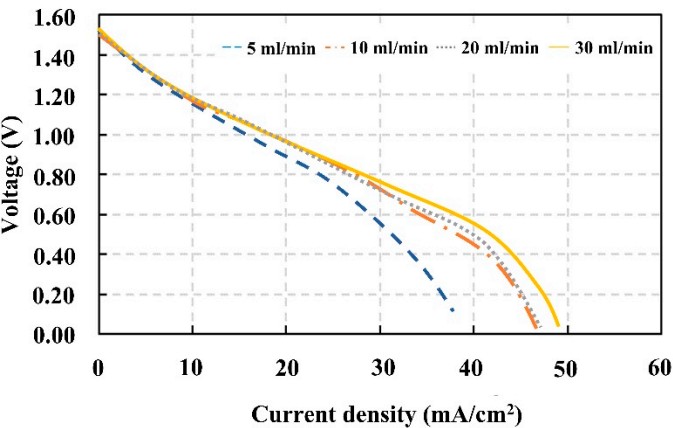

**Figure 8.** Polarization curves obtained from the discharging process after electrolyte preparation with different electrolyte flow rates.

Additionally, the electrolyte flow rate affects both the pressure and retention time of the reactant reaction in the VRFB system, as well as the electrolyte velocity. In reality, the flow rate in the VRFB system is only one factor in determining pressure; another is the flow behavior in the electrode structure. A porous electrode's fluid flow is explained by the Navier–Stokes equation. Since fluid pressure is what drives electrolyte diffusion through electrode pores, the flow rate of the electrolyte was directly influenced by fluid pressure. The permeate electrolyte through the electrode can be explained by Darcy's law [47].

Even the VRFB cell with a flow rate of 30 mL/min of electrolyte showed the best performance, with a maximum power and voltage efficiency slightly higher than the values of the cells, with flow rates of 10 and 20 mL/min. Considering the power consumption of two pumps, the two pumps used to feed the electrolytes into the VRFB cells consumed 6.91 W for a flow rate of 30 mL/min (Figure 9). That was 3.35 times the power consumed by the pumps to provide a flow rate of 10 mL/min. This demonstrates that an excessive flow rate may reduce the energy efficiency of the VRFB system. Therefore, the optimal solution flow rate for discharging was 10 mL/min.

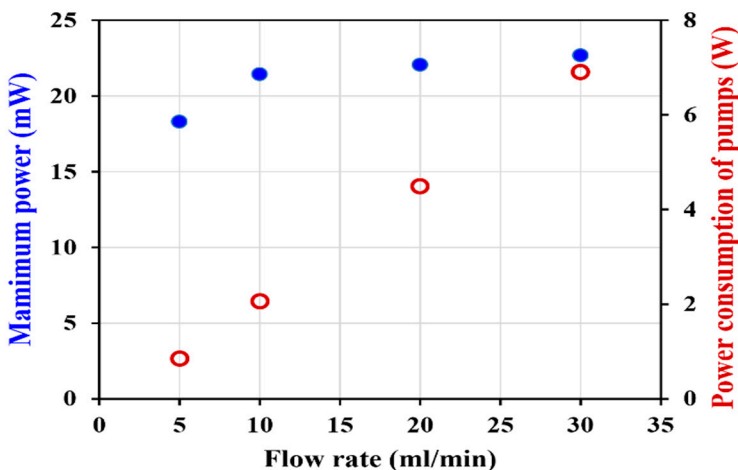

**Figure 9.** VRFB performance operated with various flow rates.

### 3.3. Observation of Applying Carbon Paper as Electrodes in VRFB

Electrodes are one of the main components of VRFBs, and they serve the function of providing an electrochemically active surface on which the redox reactions take place. When choosing an electrode, certain criteria must be met, such as being active for the chemical reaction, having a high electrical conductivity and a large surface area, being chemically stable, and being inexpensive [48]. On the surface, C–OH functional groups act as active sites for the oxidation of vanadium ions. Paper electrodes, carbon felt, and graphite felt are the electrode types used for VRFBs. In terms of electrode dimensions, there are two basic categories: 3D structure and 2D structure. While 2D electrodes such as carbon paper electrodes have historically been used for flow-by design, flow-through design frequently uses 3D electrodes such as graphite felt and carbon felt electrodes [49]. This part of the study focused on whether using carbon paper with the flow-through concept was practical. We anticipated that our flow-through architecture would work with both 3D and 2D electrode assemblies. This effort to investigate the use of carbon paper for VRFB with the flow-through design was motivated by its ease of production and economical cost. Carbon felt and graphite felts are typically made using sophisticated processes such as electrospinning followed by needle punching, whereas carbon papers are made using an electrospinning process. Carbon felt and graphite felts are three times more expensive than carbon papers. In addition, carbon paper has high electrical conductivity and high fiber density, resulting in a high specific surface area [46]. To test the impact of the number of carbon paper layers on VRFB performance, a 400 μm thick stacked carbon paper electrode was created. For this experiment, 3 and 10 layers of carbon paper were utilized as sample layers. As is well known, using multiple electrode layers will cause an increase in contact resistance between the layers [50]. In order to lower the contact resistance, the stacked electrode was compressed, which decreased the electrode's porosity and permeability. Longer charging times and a decreased reaction kinetic rate were the results of this [51], and it caused side flow surrounding the electrode, which resulted in an electrolyte shortage in the electrode's center. The concentration loss of the flow battery is closely proportional to this circumstance [46]. To achieve the desired electrical conductivity and permeability, the electrode is typically compressed in the range of 20–30% of electrode thickness with bipolar plates [51], but at high current density, more than 60 mA/cm$^2$ [52], the compression should be higher 35–50% of electrode thickness to improve electrical conductivity [53]. There are two case studies in this work; compressing three layers of carbon papers was observed in the case of 0–30% of thickness reduction, and compressing ten layers of carbon papers represented the scenario of 50% of thickness reduction. It can be assumed that the higher compressive force applied to ten-layer carbon papers meaningfully impacted electrolyte diffusion, resulting in a decrease in the reaction efficiency.

According to the electrochemical polarization results shown in Figure 9, the VRFB containing three-layer carbon papers had lower ohmic loss than the VRFB containing ten-layer electrodes. As previously stated, when the electrode layer increased, a significant voltage drop was observed at the medium current density (8.00–38.00 mA/cm$^2$). When the electrode layers increased from 3 to 10 layers, the slope of the curves in this region decreased 1.25 times. Furthermore, the power of both VRFBs supported the voltage decline phenomenon results. The maximum power of the VRFB using a three-layer electrode was 24.98 mW, while the maximum power of the VRFB using a ten-layer electrode was 21.37 mW. As previously stated, higher electrode layers increase contact resistance between electrode layers because electrons cannot transfer through conductive layers (Figure 10). The high current densities of the polarization curve for the three-layer electrode, known as the concentration overpotential region associated with a limiting current, displayed an unusual feature. The current density decreased as the cell potential decreased, owing to electrolyte depletion at a very high current density, which implies that the electrolyte ion pairs can efficiently react with each other, and the redox reaction rate of the cell with the three-layer electrode was higher than the reaction rate of the cell with the ten-layer electrode [54]. The ten-layer electrode is expected to have a highly active area, but the interpretation of the results can explain that using many layers of an electrode may reduce the porosity of the electrode, resulting in a reduction in the active area [55]. Another issue that should be investigated is electrolyte flow restriction; thus, experimental activity related to flow behavior from the flow channel through the electrode was demonstrated in this section (Figures 11–14). It should be noted that the electrolyte flowed naturally in this section of the work, with no driving pressure from a pump. The flow of electrolytes from horizontal feeding channels to a flow frame and the ten-layer electrode is depicted in Figure 11. The electrolyte did not completely flow through all ten layers of the electrode, resulting in an incomplete redox reaction. Figure 12 depicts the flow of electrolytes through an electrode in a horizontal flow battery. The electrolyte completely flowed through the flow frame and diffused into the electrode layers via a laminar flow pattern; however, the bottom four layers were discovered to be dry. Even when the electrolytes were flowing under the pump's driving pressure, the ten-layer electrode was not completely wet.

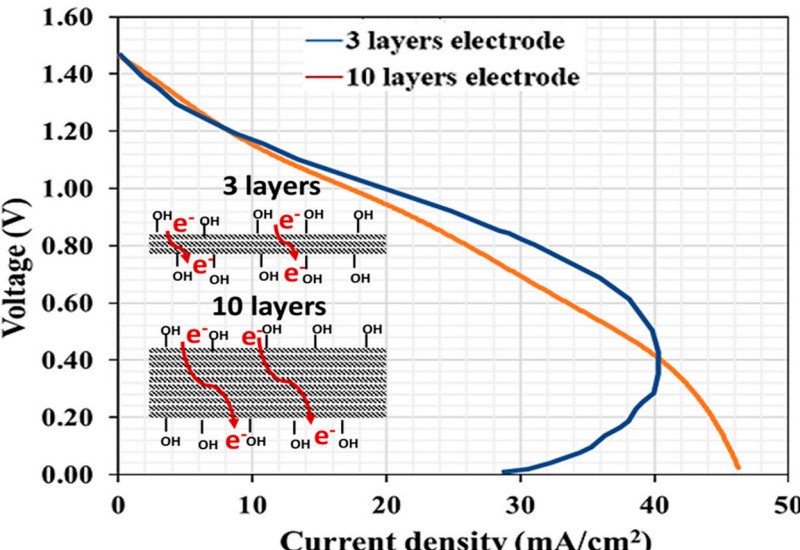

**Figure 10.** Polarization curves obtained from the discharging processes with three-layer and ten-layer electrodes.

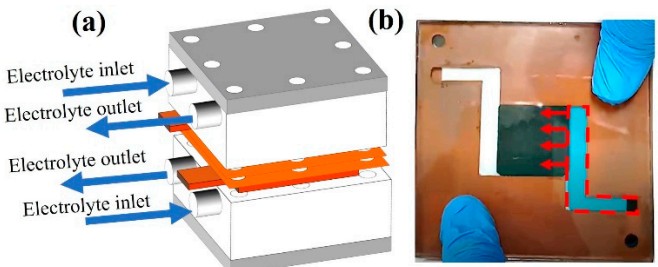

Uniform flow distribution

**Figure 11.** (**a**) Horizontal electrolyte feeding channel and (**b**) a snapshot of liquid flow into the flow frame and electrode layers.

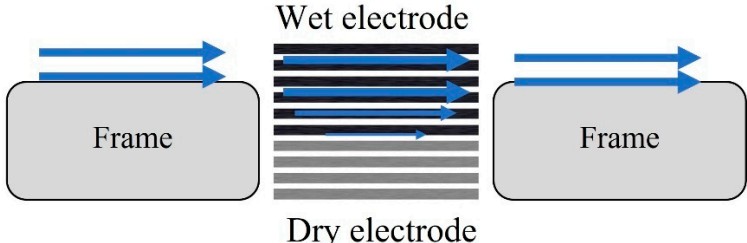

**Figure 12.** The schematic of electrolyte diffusing into electrode layers in a horizontal flow battery.

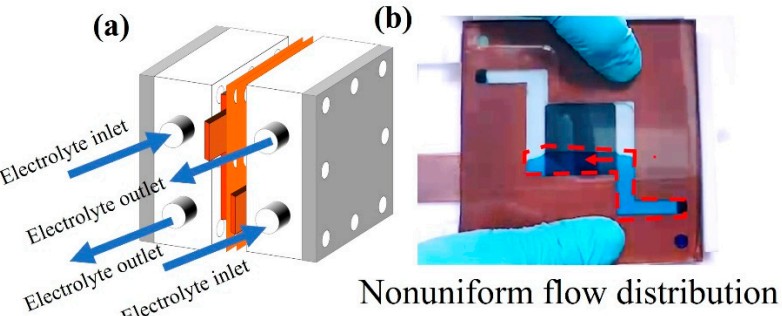

Nonuniform flow distribution

**Figure 13.** (**a**) Vertical electrolyte feeding channel and (**b**) a snapshot of liquid flow into the flow frame and electrode layers.

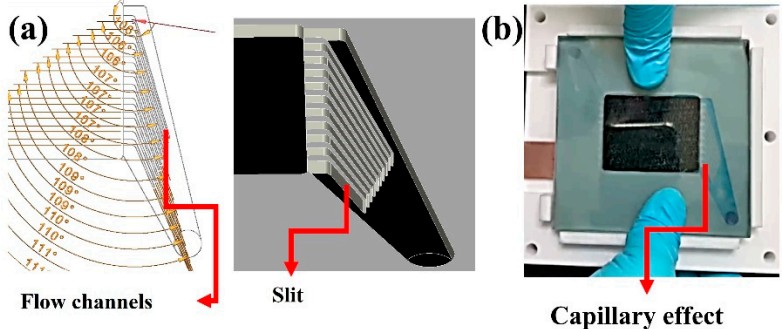

**Figure 14.** (**a**) the schematic of cone flow channels and (**b**) electrolyte flow into the flow frame with flow channels.

In the case of vertical feeding, the electrolyte diffused across electrode layers from one side to the other in a short period of time (Figure 13). The electrolyte was unable to move through all areas of the electrode, as can be seen. Shunt current is the name given to this flow pattern [31]. As a result, the horizontal flow pattern provided a greater retention time, resulting in improved electrochemical reaction performance. Another strategy for improving electrolyte flowability and diffusibility was to create cone flow channels on the flow frame.

However, this flow channel is too small, resulting in a capillary effect and high flow resistance (Equation (9)) [56]. The capillary effect can cause air to be trapped, resulting in an unstable flow (Figure 14).

$$R = \frac{8\eta L}{\pi r^4} \tag{9}$$

where $\eta$ is fluid viscosity, L is the length of the tube, r is the radius, and R is the resistance to flow.

### 3.4. The Influence of Different Nafion Membrane Specifications on VRFB Performance

In a VRFB, the cathode and anode are divided by a membrane. In order to finish the redox reaction, it also makes ion movement easier [57]. The production of protons in a vanadium redox flow battery occurs technically through two processes: the dissociation of sulfuric acid, the electrolyte's supporting medium, and the reaction of water with VOSO4 to form protons. To balance the charge on both sides for the charging and discharging processes, the generated protons then move through a membrane. Proton imbalances on both sides can lead to the generation of the vanadium cross-over. The driving forces of diffusion, migration, osmotic force, and electroosmotic convection can produce the vanadium cross-over [58]. The losses in open circuit voltage and voltage efficiency are caused by the vanadium cross-over [59,60]. A membrane's primary property requirements are good ion conductivity, high chemical resistance, a wide temperature range for application, high mechanical properties for supporting pressure generated by pumps, and good ion selectivity to protect against crossover, a problem that reduces capacity and energy.

Proton mobility through the membrane to balance charges on both sides for the charging and discharging processes is required to finish the electrochemical reactions. Therefore, to provide strong corrosion resistance, low vanadium crossover, and high proton conductivity, a membrane must be used. A cation exchange membrane, such as Nafion, has been used for VRFBs. Nafion is a sulfonated perfluorinated polymer with a copolymer molecule structure that consists of two major parts: a hydrophobic backbone (fluorocarbon), and a hydrophilic domain (sulfonate group). During the electrochemical reaction, $SO_3^-$ groups interact with water molecules by electrostatic forces in the membrane to form hydronium ions with hydrogen bonding $((H_3O)^+)$ $(SO_3H + H_2O \rightarrow SO_3^- (H_3O)^+)$. Thus, the number of sulfonate groups should influence the redox reaction capability. Theoretically, three mechanisms, namely, the surface mechanism, the vehicle mechanism, and the Grotthus mechanism, have been used to describe the proton conduction through a sulfonated membrane. Protons react with water molecules to generate hydronium ions in the case of the surface process, and they repeatedly hop on sulfonate groups on ionomer surfaces. The hydronium ions can move freely on the sulfonate group structure because the intermolecular force between sulfonate groups and hydronium ions is smaller than the molecular force between protons and sulfonate groups. In terms of the driving mechanism, protons interact with several water molecules to produce the hydronium ion. The hydronium ions migrate to balance charges through hydration and osmosis processes because the charge on both sides is unbalanced [61]. This transport process is directly influenced by the thickness of a membrane. A thick membrane has a low driving force and a high proton mobility resistance. According to the Grotthus mechanism, protons interact with water molecules to generate hydronium ions, which are then transported by hopping along the water molecules absorbed in the chemical structure of the membrane. It is worth noting that these three mechanisms were proceeded simultaneously. According to the aforementioned transport mechanism, this part of the work addressed selecting a commercial Nafion membrane for use in the VRFB, and the selection criterion was related to membrane thickness and the number of sulfonate groups. Regarding the thickness issue, Nafion 112, Nafion 1135, Nafion 115, and Nafion 117 are commercial Nafion membranes that were considered for the first selection because they contain the same number of sulfonate groups while their thicknesses are different. Nafion 112, Nafion 1135, Nafion 115, and Nafion 117 have thicknesses as follows: 50, 88, 125, and 175 μm, respectively. Research literature stated

that Nafion 117 shows superior performance than the other membranes in this group since it offers many advantages such as the greatest protection against vanadium diffusion to prevent self-discharge, the highest coulombic efficiency, the lowest swelling value (63%), and the highest value of ion exchange capacity (0.88 mmol/g) [62]. Nafion 212 and Nafion 117 were used to study the performance of the flow battery. Proton transfer in the membrane is an important factor affecting the performance of the flow battery. The thickness of the membrane and the sulfonated group affect the proton transfer in the membrane. Nafion 212 and Nafion 117 have different thicknesses and amounts of functional groups. The functionality of Nafion 117 and Nafion 212 membranes, representing widely used membranes in the market [63], for VRFB operation was investigated [64]. These Nafion membranes have sulfonate ($SO_3^-$) groups at the end of the polymer chain, which allow only protons to transfer [23]. The thickness of the membranes and the amount of $SO_3^-$ groups in them were the focus of this experiment. Serial numbers were used. Nafion membranes were labelled with the gram of polymer/mole of sulfonate group and thickness as follows: Nafion 117 means the membrane has 1100 g of polymer/mole of sulfonate group and a thickness of 0.007 inches (183.0 μm), while Nafion 212 has 2100 g of polymer/mole of sulfonate group and a thickness of 0.002 inches (50.8 μm). Note that Nafion 117 is the commercial membrane that contains the highest amount of sulfonate groups compared to the other commercial Nafion membranes.

Figure 15 shows that the VRFB containing Nafion 212 performed better than the cell containing Nafion 117. It produced open-circuit voltages of 1.51 V, a power of 47.33 mW, a maximum current density of 66.95 mA/cm$^2$, and maximum energy of 0.13 Wh. The VRFB assembled with Nafion 117, on the other hand, provided 1.48 V open-circuit voltage, 21.81 mW power, 46.26 mA/cm$^2$ maximum current density, and 0.04 Wh maximum energy. In terms of voltage loss at medium current density, the VRFB with Nafion 212 had a significantly lower ohmic loss than the VRFB with Nafion 117 [65]; thus, the material resistance was reduced. The lower resistance had a positive effect on proton transfer through the membrane that enhanced the redox kinetics [66]. As a result, it can be interpreted that the proton transfer through the membrane via vehicle and Grotthus mechanisms strongly influences on VRFB performance in comparison with the surface mechanism. As a result, this could be observed from Nafion 212, which contained a smaller number of sulfonate groups than Nafion 117; therefore, the proton transfer ability via the surface mechanism should be lower than that when Nafion 117 is used. The results can confirm that the bulk mechanism had a greater effect than the surface mechanism, and the proton transfer was coupled with the water movement via electro-osmotic drag [23]. In actuality, the performance of the cell containing Nafion 212 as shown in the polarization curves was inferior to that of the cell containing Nafion 117. When it comes to membrane thickness, Nafion 212 is less thick than Nafion 117, and it performs better than Nafion 117. This indicates that a smaller membrane thickness improves proton transfer capabilities through the Grotthus and vehicle processes [67,68]. Furthermore, the transfer resistance will decrease as the membrane thickness decreases, according to Ohm's law. Thinner membranes aid in improving proton conductivity and proton permeability. In terms of mechanical properties, Nafion 212 has a lower swelling ratio than Nafion 117 [67,68]. With this rational information, the thickness of the membrane has a greater impact than the number of sulfonate groups; thus, Nafion 212 performed better than Nafion 117.

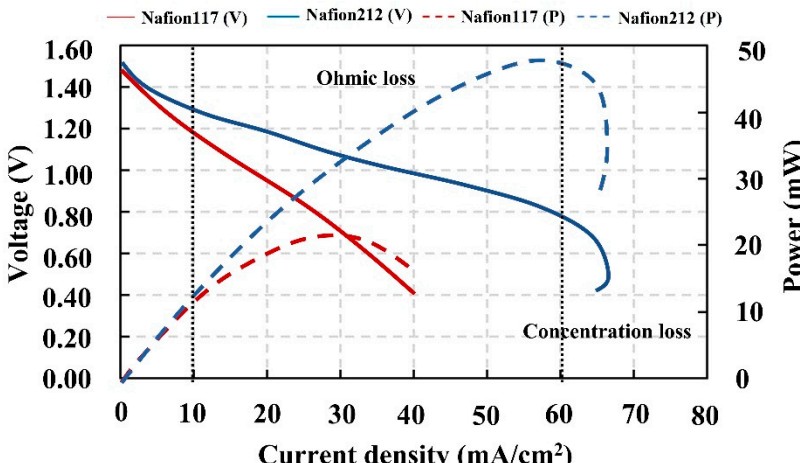

**Figure 15.** Polarization curves and power vs. current density plots of the VRFB containing various membranes.

## 4. Conclusions

Experimental results confirmed that electrochemical polarization curves can be used for observing changes in charging and discharging performances. The discussions about critical operating parameters and materials selection provide a better understanding of cell design and operating condition for further cell performance improvement. The following conclusions were extracted from this research:

(1) The highest charging performance was obtained when 1.7 V of charging voltage was imposed. Because the optimal charging voltage must overcome the standard cell potential and activation loss, this electrical potential has a significant impact on the electrochemical reactions generated during the electrolytic process. Furthermore, overpotential can cause side reactions and impurities, which can reduce charging efficiency.

(2) The VRFB, operated using electrolytes prepared by 1.7 V of charging voltages, offered the lowest ohmic loss and concentration losses. Moreover, the completion of the electrochemical reaction during initial charging directly impacts the concentration of species in the catholyte and anolyte, and it brings about the enhancement of maximum power and voltage efficiency.

(3) Even though the VRFB operated using 30 mL/min of electrolyte flow rate had the best performance, with a maximum power and voltage efficiency, 10 mL/min of the flow rate was preferred for application to the VRFB operation. Choosing the appropriate flow rate to achieve the desired performance corresponds to striking a balance between VRFB performance and pump energy efficiency.

(4) The feasibility of using a carbon paper electrode for a flow-through design was investigated in terms of the number of electrode layers as well as electrolyte flow and diffusion behavior in flow channels and electrode layers. A ten-layer electrode is anticipated to offer a highly active area, but results indicate that using many layers of an electrode may reduce the porosity of the electrode, resulting in a reduction in the active area. This phenomenon causes an increase in the redox reaction rate of the reaction generated in the VRFB assembled with the ten-layer electrode.

(5) Feeding electrolytes in different directions, horizontal and vertical, resulted in a better understanding of the effects of flow behavior on retention time and reaction capability. The horizontal flow pattern provided a greater retention time, resulting in improved electrochemical reaction performance.

(6) The thickness of the membrane and the number of sulfonate groups are vital factors to consider when selecting a sulfonated membrane for VRFB operation because they have a direct influence on the proton conduction mechanism. The performance of the cell including Nafion 212 was inferior to that of the cell consisting of Nafion 117. The

concluding result can be interpreted as follows: the thickness of the membrane has a greater effect on proton transfer than the number of sulfonate groups.

Future research will concentrate on connecting the causes to the charging kinetics in order to contribute to the overall reliability of charging VRFB. A systematic reliability analysis will be developed to gain a better understanding of how to operate VRFB with the desired performance.

**Author Contributions:** Data curation, K.O.; investigation, K.O.; methodology, K.O., R.Y. and J.G.; writing—original draft, R.Y. and S.C.; writing—review and editing, R.Y.; writing—final draft, R.Y.; visualization, R.Y.; validation, R.Y. and S.C.; funding acquisition, R.Y. and J.G.; consultant, J.G.; resources, J.G.; formal analysis, S.C. All authors have read and agreed to the published version of the manuscript.

**Funding:** This research was funded by Thailand Science Research and Innovation (TSRI), Research and Researchers for Industries (RRi) in cooperation with Thai Marine Protection Co., Ltd., grant number PHD62I0003.

**Institutional Review Board Statement:** Not applicable.

**Informed Consent Statement:** Not applicable.

**Data Availability Statement:** Not applicable.

**Acknowledgments:** The authors would like to thank Thailand Science Research and Innovation (TSRI), Research and Researchers for Industries (RRi) (grant number: PHD62I0003), and Thai Marine Protection for their financial support. We would also like to thank The Porous Materials Engineering and Analysis Lab (PMEAL), University of Waterloo, Canada, for their assistance with all experimental activities conducted at PMEAL.

**Conflicts of Interest:** The authors declare no conflict of interest.

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
