# Peer review of "Performance Analysis and Monitoring of Vanadium Redox Flow Battery via Polarization curves"

_applsci, doi:10.3390/app122211702_

Round 1

Reviewer 1 Report

Please see below.

Author Response

Dear reviewers,

Thank you very much for all kind comments from the reviewers. The comments help us to elevate quality of our manuscript and comprehension. Furthermore, all comments are very useful for our future work. Authors tried our best to revise the article related to all comments as possible as we can.

The manuscript is grammatically corrected and edited unclear sentences by a native English teacher. The certification of proofreading and grammar checking was enclosed with this submission.

Reviewer 2 Report

The authors nicely presented a study on some critical parameters related to VRFB performance charging and discharging processes. I think this work is an immersive study in the field of redox flow battery technologies. The work is within the scope of the Journal. Authors also nicely describes the number of carbon paper layers e.g., 3 and 10 influenced active areas and the diffusion of electrolytes. I think authors presented the scientific data in an excellent manner. All the data interpretation are adequately supported by the data presented. Authors nicely described the fact that how the thickness of the membrane and the sulfonated group affect the proton transfer mechanism in membrane. Therefore, I recommend publication in its present form.

Author Response

(The authors gave the same response as above.)

Reviewer 3 Report

The topic of the manuscript fits the scope of the Journal. Nonetheless, it requires certain extra efforts to improve its quality and presentation for the prestigious journal Applied Sciences. A set of comments are expounded hereafter.

In the Keywords, in this reviewer opinion, “Polarization curve” could be added.

In the Introduction, the statement about the global market size of energy storage growth (lines 31-34) should be supported by a reference.

A common practice in scientific papers consists on placing a paragraph at the end of the introductory section describing in a brief manner the structure of the rest of the manuscript. This contributes to the readability of the document and it is suggested to be included in the present case.

The achieved results are properly expounded.

Subsection 3.1 could include a flowchart to depict the described steps, in a simple but graphical manner.

A section entitled Discussion could be added to the paper in order to place text that discusses the main findings of the work, its main limitations. The contribution to the body of knowledge and novelties of the research must be clearly highlighted in this section. In fact, part of the text of the conclusions could be moved to this proposed section.

Author Response

(The authors gave the same response as above.)

Round 2

Reviewer 1 Report

DO you NOT have  properly  used  the correct past tense of the verb to flow . You kept using flew, past tense of to fly. 

Author Response

Dear reviewer,

Thank you very much for the kind comment.
